# Efficient Cytotoxicity of Recombinant Azurin in *Escherichia coli* Nissle 1917-Derived Minicells against Colon Cancer Cells

**DOI:** 10.3390/bioengineering10101188

**Published:** 2023-10-13

**Authors:** Yi Ma, Guanshu Zhu, Lan Feng, Shoujin Jiang, Qi Xiang, Jufang Wang

**Affiliations:** 1School of Biology and Biological Engineering, South China University of Technology, Guangzhou 510006, China; 2Guangdong Provincial Key Laboratory of Fermentation and Enzyme Engineering, South China University of Technology, Guangzhou 510006, China; 3Institute of Biomedicine and Guangdong Provincial Key Laboratory of Bioengineering Medicine, Jinan University, Guangzhou 510632, China

**Keywords:** minicells, protein delivery, azurin, colon cancer

## Abstract

Compared to chemical drugs, therapeutic proteins exhibit higher specificity and activity and are generally well-tolerated by the human body. However, the limitations, such as poor stability both in vivo and in vitro as well as difficulties in penetrating cell membranes, hinder their widespread application. To overcome the challenges, a highly efficient protocol was developed and implemented for the recombinant expression of the therapeutic protein azurin and secretion into minicells derived from probiotic *Escherichia coli* Nissle 1917. The novel coupled production with a delivery system of therapeutic proteins based on minicells was obtained through purification to enhance protein activity, circulation characteristics, and targeting specificity. This protein drug carrier integrates the production of carrier materials and the loading of expression proteins. The drug carrier also protects the encapsulated polypeptide drugs from enzymatic or gastric acid degradation until they are released. *Escherichia coli* Nissle 1917-derived minicells have natural targeting to colon cancer cells, low toxicity, and can accumulate for a long time after penetrating tumor tissue. This self-produced protein drug delivery system simplified the process of protein preparation, and its inhibitory effect on different types of colon cancer cells was verified by CCK-8 cytotoxicity assay, cancer cell invasion, and migration assay. This work provided a simple method to prepare minicell drug delivery systems for protein drug production and a novel approach for the transport of recombinant protein drugs.

## 1. Introduction

Colorectal cancer (CRC) is the third most common death cause of tumor diseases globally, and approximately 10,000 new cases of CRC are diagnosed each year worldwide. CRC is an inherited disease in which cumulative genetic changes in the colon confer adenomatous polyp cells with malignant properties of uncontrolled growth and the ability to invade neighboring tissues, leading to metastasis. Despite advances in prognosis and treatment, CRC remains a disease with excessive morbidity and mortality [1]. The toxicity and side effects of chemotherapy drugs limit their clinical use [2], but protein drugs have the characteristics of high activity, strong specificity, low toxicity, clear biological function, and are conducive to clinical application. At the same time, the stability of protein drugs is poor, and they are not easily absorbed, especially when given orally [3,4,5], which restricts the wide use of protein as drugs in clinical practice. Consequently, studies have focused on ways to transport and modify protein drugs to improve their efficacy [6,7].

Minicells are nano-sized bacteria that have proven to be ideal for tumor-targeted drug delivery systems to deliver chemotherapy drugs into tumors [8,9,10]. The formation of minicells is mainly through the control of bacterial division genes and the inhibition of the polar site of cell division. The systems that regulate cell division include the nucleoid occlusion system and the Min system. The bacterial Min system consists of three genes: MinC, MinD, and MinE. Mutations in the MinC or MinD genes lead to frequent segregation near the cell poles rather than in intermediate cells, followed by the formation of small spherical minicells lacking chromosomal DNA or long filamentous cells containing chromosomal DNA [11]. Similar to their parental bacterial cells, minicells also contain peptidoglycans, ribosomes, proteins, RNA, and plasmids but lack chromosomal DNA. Therefore, minicells cannot grow or divide but still maintain the activities of other cells, including ATP synthesis, mRNA translation, transcription, and plasmid DNA translation [12]. Therefore, knocking down MinC and MinD or overexpressing MinE in the bacterial genome can induce abnormal cell division and produce a large number of minicells [13].

To date, some minicells have been used to target tumors by delivering chemotherapy drugs to inhibit cancer growth. Compared with other conventional nanoparticles, minicells have several properties that make them the most suitable drug delivery tools for cancer therapy, including biocompatibility, reduced immunogenicity, lower drug leakage, reduced toxicity, higher drug loading, and therapeutic index [14]. For example, drug-loaded minicells modified with cancer cell receptor antibodies or acid ligands can selectively enter tumors and release anticancer drugs to kill cancer cells [15,16]. In recent years, due to the advantages of minicells in cancer treatment, several clinical trials have been successful in cancer patients without obvious side effects, showing that minicells are well tolerated.

Azurin is a copper-containing 128-amino acid (14 kDa) member of the ferredoxin family of REDOX proteins. The advantage of Azurin is that Azurin can preferentially penetrate cancer cells through endocytosis, caveolae body orientation, and cave body independence [17]. After preferentially penetrating cancer cells, azurin inhibits tumor cell growth through several mechanisms, leading to tumor cell shrinkage and death. The significant cytotoxicity of azurin against human cancers, such as melanoma, breast, liver, lung, prostate, ovarian and colorectal sarcomas, and fibrosarcomas, has been well studied [18,19]. Azurin acts mainly by interfering with several signaling pathways related to cancer progression. It can form a complex with the tumor suppressor protein p53 and increase the intracellular level of E3 ubiquitin ligase COP1 by inhibiting the binding of COP1 to p53 [20,21]. In addition, azurin can interfere with the autophosphorylation of tyrosine residues in the EphB2 kinase domain, thereby preventing tumor progression and inhibiting tumor growth [22,23]. Finally, azurin also inhibits tumor angiogenesis by noncompetitive inhibition of phosphorylation of vascular endothelial growth factor receptor 2 (VEGFR2) and basic fibroblast growth factor (EGFR) [24]. Each of these mechanisms of action has shown that azurin causes significant regression of several solid tumors [25,26,27].

Nonpathogenic *Escherichia coli* Nissle1917 (ECN) is a facultative anaerobic probiotic with high safety. ECN, as a facultative anaerobe, may use complex mechanisms to target tumors. It lacks significant virulence genes and exhibits adaptive factors that contribute to its colonization efficiency and survival in the host [28]. In addition, the presence of serum-sensitive lipopolysaccharide (LPS) on the ECN membrane ensures rapid clearance of the strain from normal organs and the absence of immunotoxic side effects in patients. These distinctive features are likely to shield it from elimination by the host immune system. A suitable bacterial strain is essential for effective lytic cancer therapy. The widely used serotype of salmonella enteritidis, salmonella typhimurium, has been included in phase I clinical studies for cancer therapy [29,30]. However, its tumor targeting was lower than that of Shigella flexneri 21SC602 with attenuated virulence and deletion of the ICSA gene. Several strains of Escherichia coli, including *E. coli* Nissle1917, showed the strongest tumor targeting. This may be due to the serum-sensitive LPS structure of *E. coli* Nissle 1917, which ensures rapid elimination from other organs [31]. In addition, *E. coli* Nissle 1917 has been widely used to treat acute diarrhea and some intestinal diseases in infants and young children, as well as daily health products [32]. Notably, minicells derived from *Escherichia coli* Nissle1917 have increased safety and natural targeting against colorectal cancer. At the same time, it is much better to display ligands on the surface of the minicell membrane by genetic engineering than to form non-covalent bonds between antibodies and antigens during fermentation [33]. Furthermore, the method does not require further modification. Based on these advantages, we selected *E. coli* Nissle 1917 as the host source of the drug delivery system to deliver azurin to solid cancers. Minicells have been used to treat some tumors, but there are few studies on colon cancer, and the use of minicells loaded with self-expressing therapeutic proteins in colon cancer has not been reported.

In conclusion, an ECN-derived minicell drug vector has been genetically engineered for targeted delivery of the protein drug azurin to the tumor region. Minicells were mass-produced from ECNs by deleting the MinCD gene and enhancing the expression of the MinE gene in the bacterial genome using Red/ET recombination. The protein drug azurin was expressed inside the minicells. Azurin-bearing minicells were extracted directly from the strain’s fermentation broth and used to load chemotherapy drugs.

## 2. Experimental Section

### 2.1. Materials

β-propionolactone (Maclean, Shanghai, China), Ceftriaxone (Sangon, Shanghai, China), Fetal bovine serum (FBS) (Lonsera, Canelones, Uruguay), 0.25% Trypsin-Edta, DMEM medium, Penicillin/Streptomycin, 1× PBS (GIBCO, Carlsbad, CA, USA), CCK-8 kit (Dojindo Laboratories, Kumamoto, Japan), immunostaining fixation solution, 0.25% Trypsin (no EDTA, no phenol red), apoptosis detection kit (Bi yuntian Technology Co. Ltd., Beijing, China), BD Matrigel (Corning, New York, NY USA), Transwell Chamber (BIOFIL, Guangzhou, China), RNA extraction Kit (Vazyme, Jiangsu, China), Hieff Unicon^®^ TaqMan Multiplex qPCR Master Mix (YEASEN, Shanghai, China). Bonferroni multiple comparison test (GraphPad Prism 8) was used for statistical analysis. A P value of <0.05 was considered significant. *** means *p* < 0.001, ** means *p* < 0.01, * means *p* < 0.05.

### 2.2. Methods

#### 2.2.1. Cell Cultures and Plasmid Constructs

The human colon cancer cell line HT29 was grown in DMEM supplemented with 10% FBS and 1% penicillin/streptomycin and incubated at 37 °C in a 5% CO_2_ humidified incubator. The murine-derived colon cancer cell line CT26 was grown in 1640 supplemented with 10% FBS and 1% penicillin/ streptomycin and incubated at 37 °C in a 5% CO_2_ humidified incubator.

This azurin gene fragment was cloned from Pseudomonas aeruginosa (PAO750) and cloned into Novagen T7 systems based on pET28a (Merck, Darmstadt, Germany). pET28a-AZURIN and pUC19-GFP were expressed in *Escherichia coli* Nissle 1917and BL21(DE3), purified by affinity chromatography.

#### 2.2.2. Preparation and Characterization of Minicells

Deletion of the MinCD gene and overexpression of the MinE in EcN was carried out using the Red/ET homologous recombination technique [32]. First, the pKD46 plasmid was electroporated into *Escherichia coli* Nissle 1917 to induce the expression of recombinase and screened on an Ampicillin (Amp)-containing lysogeny broth (LB) plate to obtain the positive strain (EcN harboring the pKD46) following the general protocol. The Kana gene (KANA) and ribosome binding site (RBS) were amplified from pET28a using MinCD-R1 and MinCD-F1 as primers. After purification, the amplified products were electroporated and transferred to ECN cells containing pKD46 for Red/ET homologous recombination. The obtained strains were cultured on chloramphenicol plates to obtain ECN with MinCD deletion and MinE high expression (EcN*ΔminCD::minE*). Positive EcN*ΔminCD::minE* strains were sequenced and sequenced. The gene sequences and primers used in the study are listed in the Appendix A. EcN*ΔminCD::minE* cells carrying pUC19-GFP were grown overnight in LB solution at 37 °C and transferred to fresh medium at 2% (*v*/*v*). Then, when the OD_600_ value of the culture medium reached 1.1, EcN*ΔminCD::minE* cells were harvested and washed three times with phosphate-buffered saline. The morphology and minicell germination of EcN*ΔminCD::minE* cells were photographed with an inverted fluorescence microscope (IX83). In addition, bacteria were washed three times with sterile water and morphologically observed using a field emission scanning electron microscope (MERLIN). The transformed strains were grown in LB medium supplemented with 50 μg/mL Kana resistance. Approximately 90% of the parental strains were isolated from the minicells by initial differential centrifugation at 4000× *g* for 20 min at 4 °C [34]. Then the supernatant was centrifuged at 10,000× *g* for 20 min to make minicell particles. In the next step, the pellet was re-suspended in fresh LB broth; 100 μg/mL ceftriaxone and 0.05% β-propiolactone were added and incubated at 37 °C for 2 h with shaking at 220 rpm and finally placed in a refrigerator at 4 °C for 15 min to promote cell growth and digestion of β-propiolactone. Finally, the supernatant was centrifuged at 10,000× *g* for 20 min and then filtered through a 0.45 μm filter (microwell) to finally prepare the minicells (Figure 1). At the same time, some of the harvested minicell samples were pressed onto LB AGAR plates to check for the presence of viable parental strains [35,36,37]. The morphology of purified microcell particles was also observed by transmission electron microscopy.

#### 2.2.3. SDS-PAGE and Western Blot

Individual colonies(EcN) were seeded overnight in Luria Bertani (LB) medium containing 50 μg/mL kanamycin and 100 μg/mL ampicillin (220 rpm, 37 °C). The seed solution containing 50 μg/mL kanamycin and 100 μg/mL ampicillin was expanded at 1:100 (37 °C, 220 rpm) and induced by 0.5 mM isopropyl β-d-thiogalactoside at OD_600_ = 0.8. After induction, the cells were cultured at 37 °C and 220 rpm for 6 h. Finally, the cells were collected, and the sample was prepared. The protein was expressed in Escherichia coli BL21 (DE3) as described above and then purified by Ni-NTA column and ultrafiltration tube (10 kDa). After ultrafiltration, the protein was packaged and stored at −80 °C for later use. The BCA Protein Assay Kit was used to measure the protein concentration. After centrifugation at 10,000× *g* for 2 min, the purified minicells (1 × 10^6^/mL) loaded with azurin were resuspended with 40 μL 1× PBS and mixed with 10 μL 5× loading buffer and boiled for 8 min to prepare Western blots. Image J was used to compare the band intensity of purified protein samples to estimate the protein content in minicells. 

#### 2.2.4. Extracellular Stability of Minicells_azurin_

The extracellular stability of the minicells_azurin_ was examined by monitoring changes in the particle size over 72 h using DLS. The samples were incubated in 1× PBS, DMEM cell medium + 10% FBS, or mouse plasma before being subjected to the DLS measurements. At the same time, the prepared samples were placed in pH = 7.4/6.0/4.5/2.2 solution for 24 h, and the particle size was detected. The experiments were performed in triplicate, and the results are expressed as the mean ± SD (n = 3).

#### 2.2.5. Cytotoxicity of Minicells_azurin_ against Colon Cancer Cells and IC_50_

The CT26 and HT29 colon cancer cells were seeded in a 96-well plate and cultured in an incubator overnight. Next, the culture media was replaced with different concentrations (0, 10^4^/mL, 10^6^/mL) of minicells, minicells_azurin_, or minicells_GFP_ for the cytotoxicity assay. After 24 h of incubation, 10 μL of cell counting kit-8 was added to each well of the 96-well plate and incubated for 1.5 h at 37 °C. At the same time, the culture media was replaced with different concentrations (0, 100, 200, 400, 800 μg/mL) of azurin for the cytotoxicity assay. After incubating for 24, 48, and 72 h, 10 μL of cell counting kit-8 was added to each well of the 96-well plate and incubated for 1.5 h at 37 °C.

Finally, to calculate the IC_50_ of azurin protein, after conducting the above cell plate again, separate 0, 100, 200, 400, 800, and 1600 μg/mL azurin protein and 10, 10^2^/mL, 10^4^/mL, 10^6^/mL, 10^8^/mL minicells_azurin_ replaced culture media, incubated for 24 h, and then 10 μL Cell Count Kit-8 was added to each well of the 96-well plate and incubate at 37 °C for 1.5 h. The absorption at 450 nm of each well of the plate was recorded using a Multimode Reader (Infinite M200). The respective values of IC_50_ were calculated through statistical analysis of absorbance. Minicells_azurin_ and free-azurin were also tested for the same protein concentration as minicells_azurin_ for two kinds of tumor cells.

#### 2.2.6. Assessment of Cancer Cell Apoptosis

Cancer cell apoptosis was assessed using the Annexin V Apoptosis Detection kit (Beyotime, Shanghai, China). Cells were plated in 6-well plates (BIOFIL) at a density of 5 × 10^5^ and 4 × 10^5^ cells/well for CT26 and HT29 cell lines, respectively. On the next day, the medium was exchanged for 10^6^/mL minicells_azurin_ culture media. After 24 h incubation, cells were harvested and stained for Annexin V and propidium iodide (PI) and analyzed by flow cytometry (BD FACS Aria).

#### 2.2.7. Inhibition of Migration and Invasion of Colon Cancer Cells by Minicells_azurin_

Approximately 2 × 10^5^/mL CT26 cells and HT29 cells were spread in Transwell inner chambers in 24-well plates. Minicells_azurin_ (10^6^/mL) medium containing 50% serum was added to the outer chamber and cultured in an incubator for 24 h. The next day, the liquid in the chamber was aspirated; cells were washed twice with PBS and then fixed with 4% paraformaldehyde at room temperature. After 20 min, the fixation solution was removed, and cells were washed twice with PBS and stained with 0.01% crystal violet for 15 min. Finally, gently wipe the upper layer of the inner cavity and use a microscope to select any 9 fields of view to capture the lower layer of cells. For the tumor cells invasion assay, Matrigel was melted in advance and spread on the upper layer of the Transwell chamber overnight in the cell incubator. Approximately 2 × 10^5^/mL CT26 cells or HT29 cells were cultured in the upper compartment of the Transwell while 10^6^/mL minicells_azurin_ medium containing serum was added to the 24-well plates. Incubation was held at 37 °C and 5% CO_2_ for 24 h. The subsequent processing was performed as the above migration experiment. Finally, the dye was removed, and nine images were taken randomly for each sample. The statistical data were analyzed by Image J software v1.8.0 to calculate the inhibition rate of migration and invasion.

#### 2.2.8. Cancer-Cell-Specific Uptake of Minicells

Human normal colonic epithelial cells (NCM460) and human colon cancer cells (HT29) were cultured in 6-well plates at a density of 5 × 10^5^ cells/mL for 24 h in an incubator at 37 °C, 5% CO_2_. Minicells were incubated with 150 μg/mL doxycycline (DOX) overnight at 37 °C. The next day, the medium was replaced with fresh minicells_DOX_ (10^6^/mL) or minicells. After 2 h incubation in the cell incubator, the cells were washed twice with PBS and collected. The fluorescence of DOX in the cells was recorded and analyzed by flow cytometry (BD FACS Aria). Simultaneously, NCM460 and HT29 cells were seeded on a coverslip in a 12-well plate. After incubation overnight, the medium was replaced with minicells_DOX_ (10^6^/mL) or DOX, and the cells were incubated for 2 h. Next, the cells were fixed in 4% paraformaldehyde. After washing thrice with PBS, images of the distribution of minicells_DOX_ in the cells were captured using a fluorescence microscope (OLYMPUS-IX71, Tokyo, Japan).

#### 2.2.9. The Expressions of Apoptosis-Related Genes and Tumor-Related Proteins Were Detected by Q-PCR

CT26 cells and HT29 cells were seeded in 12-well plates at a density of 3 × 10^5^ cells/mL and cultured for 24 h. Next, the medium was changed to a fresh medium containing PBS/minicells_azurin_/minicells (10^6^/mL) containing serum, and the incubation was continued overnight at 37 °C. The next day, RNA was extracted from cells under different treatments with an RNA extraction kit. After reverse transcription and Q-PCR, the regulation of p53 after protein drugs were analyzed. The primers for Q-PCR are listed in the Appendix A.

## 3. Results and Discussion

### 3.1. Construction of the Minicell for Delivery of Self-Generated Azurin to Cancer Cell

In the ECN genome, MinC and MinD genes were simultaneously deleted, and MinE gene expression was enhanced to control the site of bacterial division, thus obtaining a large number of minicells (Figure 1A–D). The fragment of azurin cloned from Pseudomonas aeruginosa was inserted into the vector to form a plasmid that could successfully express azurin. This plasmid was transferred into *E. coli* BL21(DE3) and genetically modified ECN. Expression of azurin in ECN*ΔminCD::minE* (Figure 2A,B) should permit incorporation of the protein into minicells during the minicell formation. As studied by Yu [28], proteins expressed by bacteria themselves are enriched in minicells. The expression of azurin in EcN*ΔminCD::minE* was optimized by different concentrations of inducers and the time of adding inducers. The expression of azurin in the supernatant reached a maximum when the bacteria grew to the later logarithmic stage, and 0.5 mM IPTG was used for induction. The purified protein concentration was 1653.3 μg/mL as measured by the BCA kit. Finally, the content of purified protein and protein in minicells was compared by the image intensity after the western blot. It was estimated that every 10^6^/mL minicell contained 172 μg/mL of azurin (Figure 2C). As confirmed from the dynamic light scattering (DLS) analysis, aggregation or disassembly of the minicells was not observed following incubation for 72 h in diverse media, that is, phosphate-buffered saline (PBS), medium (DMEM) with 10% fetal bovine serum (FBS), mouse plasma, and different pH solutions (Figure 2D).

### 3.2. Cancer Cell Proliferation Decreases and Cell Death Increases upon Treatment with Azurin and Minicell_azurin_

Mouse CT26 colon cancer cells and human HT29 colon cancer cells were used to investigate the in vitro cytotoxic effects of the engineered minicells and whether they have an inhibitory effect on colon cancer cell proliferation. Cytotoxicity tests showed that in colon cancer cells with different sources and types of p53, compared with the control group, both Azurin and minicell_azurin_ showed obvious inhibitory effects on the proliferation of cancer cells (Figure 3A,C). The results are presented as variation (%) in proliferation relative to the control, where no minicells were added (corresponding to a 100% proliferation rate). The effect of minicell_azurin_ seems to be inhibiting tumor cell growth, which is more pronounced with increasing concentrations. The inhibitory rate of CT26 cell activity under the maximum concentration given minicell_azurin_ reached 41.39%. At the same time, the viability inhibition rate of HT29 cells was 40.91%. Azurin had a concentration-dependent inhibitory effect on the proliferation of both types of colon cancer cells. Many studies have shown that the inhibition effect of the Celestine on mutp53-type cancer cells is not as significant as that of WT53-type cancer cells. But there was no significant difference in CT26 and HT29 cells. According to the literature reports, Celestin is the first bacterial protein found that can directly bind to the p53 protein and regulate its function. It can bind to the p53 protein in a ratio of 4:1 to form a complex [38], which can enhance the stability of the protein and improve its intracellular level, thereby enhancing the expression of the Bax gene and reducing the expression of Bcl-2. Cytochrome C is released into the cytoplasm through the mitochondrial pathway to activate caspase and induce cell apoptosis [38]. In general, minicell_azurin_ showed slightly different effects on the proliferation of the two types of colon cancer cells, but the mechanism of activity of azurin will be different for different p53 types of cancer cells. In addition, according to the image intensity comparison in the previous step, each 10^6^/mL minicells contains about 172 μg/mL of azurin. According to the comparison of Free-Azurin and minicell_azurin_ with the same protein concentration and time, the inhibition rate of minicell_azurin_ was 1.9 times higher than that of free protein for HT29 cells and 3.2 times higher than that of CT26 (Figure 3B). This is equivalent to reducing the use of drugs and increasing the absorption of drugs. As shown in Figure 3D, using GraphPad Prism 8 software to statistically analyze the IC_50_ of Azurin and minicell_azurin_, as mentioned earlier, a concentration of 10^6^/mL minicells is equivalent to Azurin with a concentration of 172 μg/mL, so 1.3 × 10^6^/mL is equivalent to 223.6 μg/mL, and 1.4 × 10^6^/mL is equivalent to 240.8 μg/mL; hence, the IC_50_ of pure proteins is significantly higher than that of minicell_azurin_. This demonstrates the stronger ability of minicell_azurin_ to inhibit tumor cell proliferation, highlighting the advantages of the minicell_azurin_ system.

According to flow cytometry analysis of the two cell apoptosis experiments (Figure 3E), the early apoptosis rate of the experimental group (minicell_azurin_) was increased compared with the control group (minicells). Moreover, the average apoptosis rate of CT26 cells and HT29 cells was increased by 2.3 and 1.3 times, respectively (Figure 3F).

### 3.3. Cancer Cell Migration and Invasion Decrease upon Treatment with Minicell_azurin_

The inhibitory effect of minicells_azurin_ on tumor cells also includes the effect on the migration and invasion of tumor cells. Minicells_azurin_ inhibitory effect on the migration and invasion of two colon cancer cells was observed by dividing it into three different chemotactic conditions: medium with minicells only and medium with no treatment. Results are given in the percentage of cancer cell migration/invasion in comparison to the control condition, where colon cancer cells were treated with only a culture medium. By observing the experimental results (Figure 4A–D), it was evident that minicell_azurin_ significantly inhibited the migration and invasion of two different types of colon cancer cells. As for the migration experiment, the mobility of the minicells_azurin_ treatment group of CT26 cells decreased by about 49.6% compared with the control group, while the mobility of HT29 cells decreased by about 25.5%. As for the invasion experiment, the invasion rate of the minicells_azurin_ treatment group of CT26 cells decreased by about 34.2% compared with the control group, while that of HT29 cells decreased by about 57.9%. In addition, by comparing the inhibition rates of migration and invasion of different cells, it can be seen that minicells_azurin_ was more effective in inhibiting the invasion of HT29 cells than CT26 cells.

### 3.4. Minicells with Tumor Tropism Promote Internalization

To verify the ability of minicells to penetrate tumor cells and promote drug internalization, minicells_DOX_ were incubated with HT29 colon cancer cells and NCM460 colon epithelial cells. The cellular uptake and intracellular distribution of DOX were visualized using a fluorescence microscope. Figure 5A clearly shows that the NCM460 colon epithelial cells display weak fluorescence signals (red color) after 2 h of incubation with the minicells_DOX_. However, minicells_DOX_-treated HT29 cells showed much stronger fluorescence than the other groups. In conclusion, the fluorescence intensity is related to the internalization of minicells, suggesting that minicells have a certain ability to target cancer cells and promote the internalization of drugs into cells. Additionally, as shown in Figure 5B, to determine whether the DOX carried by minicells_DOX_ could successfully accumulate in different cancer cells, flow cytometry was used to analyze the fluorescence intensity of the chemotherapeutic drugs in the cells. After 2 h of incubation, the rate of fluorescence positivity of minicells_DOX_-treated groups was 26.2% and 16.1% in HT29 cells and NCM460 cells. And simultaneously, the minicells_DOX_-treated HT29 cells exhibited a 1.3-fold higher fluorescence intensity than the minicells_DOX_-treated NCM460 cells (Figure 5C). The results further confirmed that minicells could efficiently drive the entry of DOX into cancer cells, which is potentially useful for targeting tumors in vivo for anticancer studies.

### 3.5. Secretion of Cytokines Involved in Different Kinds of Colon Cancers by Engineered Minicells_azurin_ Quantified by Q-PCR

To investigate the inhibitory effect of minicells _azurin_ on different types of colon cancer cells, we evaluated the expression of cytokines expressed by minicells_azurin_ that have been described to have a role in minicells_azurin_ interaction with cancer cell p53. In the end, we analyzed the concentration of the factor in the cells by Q-PCR after being treated with minicells_azurin_, minicells, and PBS (Figure 6). The results are given as the relative fold change in p53 expression relative to the control condition (without minicells_azurin_ or minicells). We observed that in CT26 cells, the mRNA level of p53 declined to about 0.25-fold of control; at the same time, in HT29 cells, p53 declined to about 0.17-fold of control. At present, it is known that CT26 cells and HT29 cells are not wild-type p53 protein-type cancer cells, but minicells_azurin_ can significantly inhibit the two kinds of cells. Through a Q-PCR experiment, it was found that the p53 protein decreased at the mRNA level, but the specific mechanism of action is still a question worth exploring.

## 4. Conclusions

In summary, a drug delivery platform for the self-production of the therapeutic protein Azurin was developed and applied against Colon cancer cells. Minicells protect proteins under acidic conditions so that therapeutic proteins can be transported within the cell and maintain their biological activity. Using this simple and low-cost strategy to transport proteins is promising for a variety of therapeutic applications. Oral treatment of peptides or proteins remains a challenge for the pharmaceutical industry and researchers due to many human conditions, such as high proteolytic and low pH conditions in the gastrointestinal tract, low permeability of proteins/peptides to the intestinal barrier, etc. Therefore, peptides require nanocarriers for modification and encapsulation. If successful, the frequency of administration is reduced, treatment costs are reduced, and patient compliance is increased. Minicells are produced by bacteria, and bacteria can simultaneously introduce plasmids that produce different proteins. Proteins produced by bacteria also aggregate in minicells [28]. At the same time, minicells could be loaded with chemotherapeutic drugs and possibly be delivered in combination with protein drugs for synergistic effects.

## Data Availability

The raw data supporting the conclusion of this article will be made available by the authors without undue reservation.

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
