# Peer review of "Efficient Cytotoxicity of Recombinant Azurin in Escherichia coli Nissle 1917-Derived Minicells against Colon Cancer Cells"

_bioengineering, 2023, doi:10.3390/bioengineering10101188_

Round 1
Reviewer 1 Report
It will be nice to explain whether protein containing disulfide bonds can be expressed and delivered. Please also mention how much of cell density of E.coli can be achieved with such strain. Higher the cell OD, hopefully high amount of mini cells will be produced and will be good for application. Does this min icells have residual endotoxin and need elaboarate protocol or removal before carrying out in vivo experiment. This will help in strenghthening the manuscript and the technology for future.
Author Response
Dear Professor,
Thank you very much for your comments about our submitted paper “Efficient Cytotoxicity of Recombinant Azurin in Escherichia coli Nissle 1917 Derived Minicells Against Colon Cancer Cells” (Manuscript Number: bioengineering-2445693). According to your recommendations, we have carefully revised our manuscript with red font. The following is a detailed list of response to all comments and criticisms. If you have any questions about this paper, please don’t hesitate to let me know.
Thank you very much for your consideration.
Most sincerely,
Yi Ma
It will be nice to explain whether protein containing disulfide bonds can be expressed and delivered. Please also mention how much of cell density of E.coli can be achieved with such strain. Higher the cell OD, hopefully high amount of mini cells will be produced and will be good for application. Does this minicells have residual endotoxin and need elaboarate protocol or removal before carrying out in vivo experiment. This will help in strenghthening the manuscript and the technology for future.
Response: Thank you very much for your advice. We have revised them in modified manuscript and added the content with red font.
There are the detailed responses:
Firstly, this article did not explore how much concentration of Escherichia coli is required to produce how many minicells. Instead, it referred to multiple literature and the conditions required for self expression of proteins to specify a culture condition. Secondly, does this minicells have residual endotoxin and need elaboarate protocol or removal before carrying out in vivo experiment? The host bacterium selected in this manuscript is Escherichia coli nissle1917 (EcN), a probiotic bacterium belongs to serum O6: K5: H1 and its LPS expresses semi rough O6 antigen. Since the polymerase wzy gene on the antigen terminates point mutation, its side chain is composed of only single oligosaccharide repeat unit and its K5 clip membrane is serosensitive, which makes EcN easy to be cleared when encountering serum in vivo. It is one of the characteristics that sets it apart from other Escherichia coli. EcN lacks pathogenic factors commonly presented in its homologous pathogenic strains and does not carry virulence factors, therefore EcN is endotoxin-free probiotics. Minicells of EcN possess this advantage, so there is no endotoxins on the human body. According to current research about minicells, drug loaded minicells have entered clinical trials and their biosafety is relatively high.
Reviewer 2 Report
In the current manuscript, authors developed a Minicell-based delivery system and analyzed its potential for cancer thera. Authors performed experiments to evaluate the Minicell system from EColi and observed cytotoxic effects on CT26 & HT29 cell lines. Manuscript lacks some key points for studying developed delivery system and didn't use any control for most of the experiments.
I have following suggestions to improve the manuscript.
Authors should use Azurin and MiniCell (Balnk formulation) to evaluate their Minicell-Azurin system.
In the figure 3A authors reported cytotoxicity data from cells using Azurin and M Azurin but authors didn’t mention the concentration of Azurin used for this experiment. I think authors should use 5-6 Azurin concentrations to study cytotoxic effects on cells and provide IC50 for Azurin and M-Azurin.
Figure 3C&D Why plain Azurin data is missing from the Apoptosis assay?
Authors should provide the amount of Azurin present in developed Minicells, as Azurin is the only active constituent in the developed system so it will be helpful to have some idea about the amount of Azurin in the formulation.
Figure 5 A Confocal images of HT29 cells treated with Minicell-DOX, why the background of merged images is different in cells with cell uptake of Minicell. The darker background is clearly visible in cells that have red signal while other cells have bright background.
Figure 5C authors should provide MFI of both the control cells.
Author Response
Dear Professor,
Thank you very much for your comments about our submitted paper “Efficient Cytotoxicity of Recombinant Azurin in Escherichia coli Nissle 1917 Derived Minicells Against Colon Cancer Cells” (Manuscript Number: bioengineering-2445693). According to your recommendations, we have carefully revised our manuscript with red font. The following is a detailed list of response to all comments and criticisms. If you have any questions about this paper, please don’t hesitate to let me know.
Thank you very much for your consideration.
Most sincerely,
Yi Ma
1. Authors should use Azurin and MiniCell (Balnk formulation) to evaluate their Minicell-Azurin system. Response: Thank you very much for your advice. We have revised them in modified manuscript with red font.
2. In the figure 3A authors reported cytotoxicity data from cells using Azurin and M Azurin but authors didn’t mention the concentration of Azurin used for this experiment. I think authors should use 5-6 Azurin concentrations to study cytotoxic effects on cells and provide IC50 for Azurin and M-Azurin.
Response: Thank you very much for your advice. We have revised them in modified manuscript with red font. We have supplemented the cytotoxicity test of azurin and calculated the IC50 values of azurin and Mincellsazurin. Due to the main purpose of this article to verify and explore the feasibility of the Mincellsazurin system, its therapeutic effect was not compared with pure azurin.
3. Figure 3C&D Why plain Azurin data is missing from the Apoptosis assay?Response: Thank you very much for your advice. We have revised them in modified manuscript with red font. Our manuscript is only to preliminarily verify whether the Mincellsazurin system has a proapoptotic effect, so only minicells are used as a control. In addition, according to multiple literature studies, the apoptotic effect of pure azurin has been fully studied, and it does indeed have a proapoptotic effect on tumor cells. In addition, as mentioned in Article 3.2, minicells with a concentration of 50% represent cells containing 106/ml of minicells, which contain a protein concentration of 172μg/ml.
4. Authors should provide the amount of Azurin present in developed Minicells, as Azurin is the only active constituent in the developed system so it will be helpful to have some idea about the amount of Azurin in the formulation.
Response: Thank you very much for your advice. We have revised them in modified manuscript with red font.
5. Figure 5 A Confocal images of HT29 cells treated with Minicell-DOX, why the background of merged images is different in cells with cell uptake of Minicell. The darker background is clearly visible in cells that have red signal while other cells have bright background.
Response: Thank you very much for your advice. We have replaced it with a new image. The different background colors of the merged image may have initially been reduced in brightness to facilitate fluorescence observation.
6. Figure 5C authors should provide MFI of both the control cells.
Response: Thank you very much for your advice. We have added the MFI values for the control group and redrawn the chart.